# Demographic and Socioeconomic Predictors of Prehypertension and Hypertension in the Adult Population: Serbian National Health Survey

**DOI:** 10.3390/medicina60050824

**Published:** 2024-05-16

**Authors:** Igor Dimitrijev, Snezana Radovanovic, Zoran Vesic, Goran Colakovic, Viktor Selakovic, Ana Lackovic, Slavica S. Djordjevic, Maja Pesic, Danijela Nesovic, Radomir Lazarevic, Ognjen Djordjevic, Olgica Mihaljevic, Aleksandra Obradovic, Verica Vukicevic, Nikoleta Janicijevic, Jovana Radovanovic

**Affiliations:** 1Department of the High School of Health, Academy of Applied Studies Belgrade, 11000 Belgrade, Serbia; igordimitrijev@yahoo.com (I.D.); slavica.djordjevic30@gmail.com (S.S.D.); 2Department of Social Medicine, Faculty of Medical Sciences, University of Kragujevac, 34000 Kragujevac, Serbia; 3Institute for Public Health Kragujevac, 34000 Kragujevac, Serbia; 4Center for Research on Harmfull Effects of Biological and Chemical Hazards, Faculty of Medical Sciences, University of Kragujevac, 34000 Kragujevac, Serbia; 5Faculty of Political Sciences, University of Belgrade, 11000 Belgrade, Serbia; zvesic@gmail.com; 6Institute for Emergency Medicine of Belgrade, 11000 Belgrade, Serbia; colakovic.goran@gmail.com (G.C.); aleksandra.obradovic.saska@gmail.com (A.O.); verica.vukicevic@gmail.com (V.V.); 7Faculty of Medical Sciences, University of Kragujevac, 34000 Kragujevac, Serbia; selakovicviktor@gmail.com (V.S.); maja.pesic96@yahoo.com (M.P.); radovanovicjovanaaa@gmail.com (J.R.); 8Health Center “Dr Milutin Ivković” Palilula, 11000 Belgrade, Serbia; lackovicmd@gmail.com; 9Clinical Hospital Center Zemun, Surgery Clinic, Baromedicine Department, 11000 Belgrade, Serbia; d.krtinic@yahoo.com; 10Dom Zdravlja Barajevo, 11000 Belgrade, Serbia; dr.radomir.lazarevic@gmail.com; 11Department of Epidemiology, Faculty of Medical Sciences, University of Kragujevac, 34000 Kragujevac, Serbia; ognjendjordjevic763@gmail.com; 12Department of Pathophysiology, Faculty of Medical Sciences, University of Kragujevac, 34000 Kragujevac, Serbia; vrndic07@yahoo.com; 13Department of Hygiene and Ecology, Faculty of Medical Sciences, University of Kragujevac, 34000 Kragujevac, Serbia; nikoleta.janicijevic@gmail.com

**Keywords:** prehypertension, hypertension, sociodemographic predictors, national health survey, Serbia

## Abstract

*Background and Objectives*: Prehypertension and hypertension are the most common cardiovascular disorders worldwide and are increasingly considered one of the most serious public health problems, particularly in developing countries. The objective of this study was to determine the frequency and demographic and socioeconomic predictors of prehypertension and hypertension in the adults in Serbia, and to examine the relationship between prehypertension and hypertension and health behavior determinants (smoking, alcohol use, physical activity) and individual aspects of health (a health self-assessment, multimorbidity, BMI, depressive symptoms). *Materials and Methods*: The research is part of the fourth National Population Health Survey conducted in 2019, which was conducted by the Republic Institute of Statistics, in cooperation with the Institute of Public Health of Serbia and the Ministry of Health of the Republic of Serbia. As a research instrument, questionnaires were used in accordance with the methodology of the European Health Survey. For the purposes of this research, data on the adult population aged 20 and over were used. *Results*: Women are at a reduced risk for both prehypertension (OR  =  0.328) and hypertension (OR = 0.349) by nearly 70%. Similarly, those aged below 60 years have a lower risk for prehypertension and those younger than 40 years have a lower risk for hypertension (OR = 0.995), whereas people with a lower education have a 4.3 times higher risk of prehypertension (OR = 4.323) and a 1.6 times higher risk of hypertension (OR =  1.614). The poor have a 1.4 times higher risk of prehypertension (OR = 1.413) and a 1 times higher risk of hypertension (OR = 1.035). People with multimorbidity have a 1.2 times higher risk of both prehypertension (OR = 1.218) and a 4.8 times higher risk of hypertension (OR = 4.867). *Conclusions*: Male gender, lower education, poverty, age and the presence of multimorbidity are significant predictors of prehypertension and hypertension in the Serbian adult population, so preventive strategies should be aimed at these sensitive population groups.

## 1. Introduction

Cardiovascular diseases are the leading cause of morbidity and mortality worldwide and the global burden of disease [1,2]. Prehypertension and hypertension are the most common cardiovascular disorders worldwide and are increasingly considered one of the most serious public health problems, particularly in developing countries [3,4].

Prehypertension is a precursor of hypertension and has a strong positive linear relationship with morbidity and mortality from cardiovascular diseases [5,6]. Prehypertension leads to hypertension, and the proportion of the young population with prehypertension is increasing, especially in low- and middle-income countries [7]. Globally, hypertension affects approximately one in four adults, and by the year of 2025 the number of adults with hypertension is projected to increase by about 60% to a total of 1.56 billion inhabitants [8].

In addition, epidemiological studies have shown that prehypertension is a common condition worldwide in as many as 30 to 50% of the studied population [9]. Approximately 90% of individuals with prehypertension have at least one cardiovascular risk factor, and 68% have at least one clinically significant risk factor for heart disease or stroke [10].

Studies have shown that the potential risk factors associated with arterial hypertension are low education, a long period of smoking [11], sex, age, place of residence (urban versus rural), psychosocial risk factors (socioeconomic status, stress), overweight status (BMI: 25–29.9 kg/m), general obesity (BMI: ≥30 kg/m), abdominal obesity (waist circumference: ≥94 cm) and diabetes mellitus [12]. 

Additionally, many people are diagnosed with hypertension accidentally or after serious organ damage due to its asymptomatic nature. The level of awareness, treatment and control of the disease is still low in underdeveloped and developing countries, including our country, and as a result, most people affected by hypertension are unaware of their status [10].

A comprehensive study on the prevalence and risk factors associated with prehypertension and hypertension in Serbia is rare. The latest available data are from the third National Health Survey in Serbia from 2013, when (17.7%) of the population of Serbia, aged 15 and over, was normotensive, and every third (33.1%) person had prehypertension, and every second (49.3%) hypertension [2].

In this context, the goal of our research was to determine the frequency and demographic and socioeconomic predictors of prehypertension and hypertension in the adult population of the Republic of Serbia, using data from the last National Health Survey of Serbia conducted in 2019. We also wanted to examine the relationship between prehypertension and hypertension and health behavior determinants (smoking, alcohol use, physical activity) and individual aspects of health (a health self-assessment, multimorbidity, BMI, depressive symptoms).

## 2. Materials and Methods

### 2.1. Type of Study

The study represents a secondary analysis of data obtained from the fourth National Health Survey of the Republic of Serbia, which was conducted by the Ministry of Health of the Republic of Serbia in accordance with the recommendations for the implementation of the European Health Survey [13] for 3 months (October–December) in 2019.

### 2.2. Target Population

The primary target population consisted of all persons aged 15 and over living in private (non-institutional) households in the Republic of Serbia, who represent the usual population. The excluded persons were the ones living in collective households (student dormitories, dormitories for children and youth with disabilities, homes for socially endangered children, retirement homes for seniors, care homes for the elderly and infirm, adult disability homes, monasteries, convents, etc.). Stratification was performed according to the type of area (urban and other) and the four regions: the Belgrade region, the Vojvodina region, the Sumadija and Western Serbia region, and the Southern and Eastern Serbia region [13].

### 2.3. Domains of Research and Stratification

The main goal of the research was to obtain statistically reliable estimates for most indicators, both at the national level and for urban and other settlements, as well as for four regions: the Belgrade region, the Vojvodina region, the Sumadija and Western Serbia region, and the Southern and Eastern Serbia region. In the research, a stratified two-stage sample was applied, which is stratified according to settlements and regions (the region of Šumadija and West Serbia, the region of South and East Serbia, the region of Vojvodina and the Belgrade region). The data of the 2011 population census were used as a framework for selecting the sample.

### 2.4. Sample Size and Sample Allocation

The sample size was calculated on the basis of the requirements related to the precision of estimates, in order to assess the standard error of the indicator “proportion of persons prevented from engaging in daily activities” in accordance with EUROSTAT recommendations for conducting the European Health Interview Survey [13]. In EU countries, based on the SILK survey, the proportion of people who are prevented from performing daily activities in the population varies between 4% and 11%. Similar results were obtained in the SILK survey in the Republic of Serbia (5–8%). If 8% is taken as the basis for further calculations, in order to estimate the parameter with an error of less than 1%, it is necessary for a simple random sample to have about 6000 subjects aged 15 and over. In that case, if the statistic has a value of 8%, the parameter estimate is in the interval between 7.3% and 8.7% (95% confidence interval). It is planned to obtain statistically reliable estimates at the level of Serbia as a whole, then at the level of the four regions: the Belgrade region, the Vojvodina region, the Sumadija and Western Serbia region, and the Southern and Eastern Serbia region, as well as for the population of urban and other settlements. As a compromise between the required precision of estimates and the cost of conducting the research, a sample size of 6000 households was determined, in which approximately 15,000 members were aged 15 and over whereas about 1500 children were aged from 5 to 14. When calculating the sample size, children aged from 5 to 14 years were not taken into account. It was determined that 10 households should be selected in each census round, taking into account the costs of conducting the survey, as well as the time required to complete the survey in the census round. Reserve households were also provided for each census round in the case that a large number of households in the census round refused to cooperate. Dividing the total number of households by the number of households in the sample per census district, it was calculated that 600 census districts should be selected. Sample allocation by region and type of settlement was proportional to the number of persons aged 15 and over in these contingents, based on the current 2018 demographic estimates. A sample of 5114 households counted 15,621 persons, out of which there were 13,589 persons aged 15 and over and 1493 children aged 5 to 14. For the purpose of this study, data on the adult population aged 20 years and older were used.

### 2.5. Sample Selection Frame and Sample Selection

The 2011 census of the Republic of Serbia was used as a sampling framework. Census circles were formed for the purpose of conducting the census, where they were defined as primary sample units and were selected from each stratum systematically with a probability of choice proportional to size, and the size measure was the number of households in each census circle based on the 2011 census. Within each stratum, the census districts were sorted according to the municipality to which they belonged and the ordinal number within the municipality. Households within each census round were selected with equal probability (simple random sampling) from the list of households recorded in the 2011 census.

### 2.6. Respondents’ Participation in the Research and Response Rate

A total of 5114 households and 15,621 respondents were included in the survey. Out of a total of 6335 contacted households, 5114 of them agreed to participate in the research. The response rate of the households was 80.7%. Out of a total of 13,589 registered members of households aged 15 and over, 13,178 of them agreed or was able to be surveyed, which gives a response rate of 97.0%. Out of the number of people who agreed to be interviewed, 11,790 agreed to fill in the self-complete questionnaire (response rate 89.5%), while 11,474 agreed to have their measurements taken (response rate 87.1%).

### 2.7. Ethical and Legal Aspects

The research participants were provided with a written document containing the necessary information related to the purpose of the study, the scope of their rights and a phone number intended for additional information or possible complaints. An informed notice was acquired via the written signature from every participant that had agreed to take part in the study. Anonymity in the study was ensured in accordance with the Law on Official Statistics, whereas the necessary identification was removed and replaced with a code).

### 2.8. Research Instruments

A household info panel, which was used to collect information about all members of the household, i.e., socio-economic characteristics of the household itself;A self-completion questionnaire, which was filled in independently by each member of the household aged 15 and over.

### 2.9. Variables

The independent variables encompassed demographic attributes such as gender, age, marital status, and region, complete with socioeconomic factors such as education, employment status, welfare index, determinants and health behavior (smoking, alcohol use, physical activity) and individual aspects of health (self-assessment of health, multimorbidity, BMI, depressive symptoms). On the other hand, the dependent variables of interest were prehypertension and hypertension (the result of objective blood pressure measurement).

Arterial blood pressure was measured in all subjects, except for those who had both arms amputated, casts on both arms, open wounds on both arms, rashes on both arms, malformations that did not allow cuffs to be placed, or lymph stagnation on both arms that prevented proper cuff placement. Blood pressure was measured using a three-cuff digital sphygmomanometer with a 230VRiChampionN adapter. The measurement was performed on the right hand, if it was possible. The measurement was performed in a sitting position, so that the arm and back were resting (on the table or the back of the chair), and the legs were touching the floor. Blood pressure was measured three times at one-minute intervals, with the systolic and diastolic pressures recorded in mmHg. The systolic and diastolic blood pressure values from the last two measurements were used to calculate the average blood pressure value. To determine the existence of elevated blood pressure (arterial hypertension) and potential arterial hypertension, data on systolic and diastolic blood pressure obtained using measurement were used, as well as information on whether a person was taking blood pressure lowering medication.

Keeping the goal of the research in mind, in which the blood pressure value was determined as the main dependent variable, three subgroups of respondents were separated within the study population: (1) the normotensive group (systolic TA < 120 mmHg and/or diastolic TA < 80 mmHg), (2) the prehypertensive group (systolic TA = 120–139 mmHg and/or diastolic TA = 80–89 mmHg), and (3) the hypertensive group (systolic TA ≥ 140 mmHg and/or diastolic TA  ≥  90 mmHg), [2,13].

### 2.10. Statistical Methods

The statistical analysis was performed using a commercial standard software package, namely the Statistical Package for Social Sciences (SPSS) software version 19.0 (SPSS Inc., version 19.0, Chicago, IL, USA). An χ^2^ test was applied to test the difference in the frequency of categorical variables. A logistic regression analysis was applied to examine the factors associated with prehypertension and hypertension. All results with the probability that was equal to or less than 5% (*p*  ≤  0.05) were considered statistically significant.

## 3. Results

The research included 5515 (52.39%) female respondents and 5010 (47.61%) male respondents. The average age of the entire study population was 52.83 ± 17.69 years (min 20, max 99 years). There were 14.71% of normotensive subjects, 49.07% of subjects with prehypertension and 36.21% of subjects with hypertension (*p* < 0.001).

Table 1 shows the differences between the three tested groups of respondents in terms of their sociodemographic characteristics and health status. In terms of gender distribution, it was shown that prehypertension (50.63%) and hypertension (52.14%) were present in a higher percentage in male subjects. The highest percentage of respondents with prehypertension was aged 40–49 (18.7%), whereas the highest percentage of respondents with hypertension was aged 60–69 (27.98%). The most common occurrence of prehypertension (64.42%) and hypertension (65.87%) was recorded in persons living in a community (marriage or cohabitation). Middle-educated subjects more often had prehypertension (55.88%) and hypertension (50.72%). The region of Sumadija and Western Serbia demonstrated a higher prevalence of prehypertension (38.43%) and hypertension (32.43%) compared to more urban parts of Serbia such as Belgrade and Vojvodina. Additionally, prehypertension (40.17%) and hypertension (45.52%) were more common in poorer people. Also, employed respondents were more frequently reported to have prehypertension (43.84%), whereas those who were inactive more often had hypertension (59.80%). The largest percentage of respondents with prehypertension (69%) and hypertension (44%) claimed to have poor or very poor general health. People with hypertension highlighted the presence of multimorbidity (52.21%). In relation to the body mass index, the largest percentage of respondents with prehypertension (39.55%) and hypertension (38.41%) belonged to the overweight category. Both respondents with prehypertension (79.80%) and those with hypertension (67.35%) reported mild depression in the highest percentage as well as the presence of depressive episodes (19.37% with prehypertension and 31.25% with hypertension); see Table 1.

Our research showed that the highest percentage of people with prehypertension (21.20%) and hypertension (20.78%) were consuming cigarettes on a daily basis. Concerning the aspect of physical activity, prehypertension and hypertension were predominant in persons who were physically active for less than 150 min (79.51% with prehypertension and 79.9% with hypertension); see Table 2.

Table 3 and Table 4 show the predictive significance of the respective socio-demographic characteristics, health status and health behavior of the respondents as regards to the occurrence of prehypertension and hypertension. Women are at a reduced risk to have prehypertension (OR  =  0.328) or hypertension (OR = 0.349) by nearly 70%. When it comes to marital status, it can be concluded that divorced people are 1.6 times more likely to have prehypertension (OR = 1.667), while never married subjects have a reduced risk for both prehypertension (OR =  0.286) and hypertension (OR = 0.761) by 70%, i.e., 20%. All age groups under 60 years have a significantly lower risk of prehypertension and those younger than 40 years have a significantly lower risk of hypertension compared to the oldest population. Residents of Šumadija and western Serbia show a reduced risk of prehypertension (OR = 0.610) by 40% and a reduced risk for hypertension (OR = 0.804) by 20%, whereas people with a lower education have a 4.3 times higher risk of prehypertension (OR = 4.323) and a 1.6 times higher risk of hypertension (OR = 1.614). The poor have a 1.4 times higher risk of prehypertension (OR = 1.413) and a 1 times higher risk of hypertension (OR = 1.035) and people with multimorbidity have a 1.2 times higher risk of both prehypertension (OR = 1.218) and 4.8 times of hypertension (OR = 4.867), whereas alcohol consumption increases the risk of prehypertension by 1.6 times (OR = 1.604) and 2.7 times for hypertension (OR = 2.740). Adults who engage in sports less than 150 min a week have a 1.2 times higher chance of prehypertension (OR = 1.213) and a 3.4 times higher chance of hypertension (OR = 3.437).

## 4. Discussion

Our results show that 49.07% of the subjects had prehypertension and 36.21% of the subjects had hypertension. In terms of gender distribution, it was shown that prehypertension and hypertension were present in a higher percentage in male subjects. In some studies, no association was found between the proportion of prehypertension and hypertension in relation to gender [14]. In contrast, other studies like ours have found a higher average prevalence of blood pressure in men than in women [15,16]. Although hypertension is conventionally more common among the elderly, recent epidemiological studies have revealed that hypertension and prehypertension can occur in young adulthood [17]. This is supported by our data, where the risk of prehypertension and hypertension was higher in younger age groups. Furthermore, there are authors who state that marriage is a risk factor for prehypertension. Hypertension was found to be positively correlated with marital status [18] which is in accordance with our research. Some authors have found that a lower level of education is significantly associated with prehypertension [19], which is also confirmed by our results, where middle-educated subjects more often had prehypertension and hypertension.

Several studies [20,21] noted that people living in urban areas are exposed to a higher risk of hypertension. This could be interpreted in terms of lifestyle changes occurring in these urban areas, including dietary habits such as easy access to fast food and the availability of transportation, all of which contributed to the increased level of physical inactivity. Other studies have established a connection with rural areas. This is most likely due to the lack of screening as a result of limited health infrastructure and low levels of education reported in these areas [22]. In our research, the region of Sumadija and Western Serbia demonstrated a higher prevalence of prehypertension (38.43%) and hypertension (32.43%) compared to more urban parts of Serbia such as Belgrade and Vojvodina. Additionally, our results showed that both prehypertension and hypertension were more common in poorer people. However, there are also studies that have not found evidence of the association between hypertension and education, income or occupation [23]. In our study, employed respondents were more frequently reported to have prehypertension, whereas those who were inactive more often had hypertension.

Many studies have shown that BMI and obesity are both correlated with hypertension [24,25,26], which coincides with our results where the highest percentage of subjects with hypertension belongs to the overweight category.

Numerous studies have observed a correlation between prehypertension and people’s lifestyle. A sedentary lifestyle, low physical activity, smoking and alcohol consumption, and low fruit and vegetable consumption were associated with the occurrence of prehypertension and hypertension [22]. Our research showed that the highest percentage of people with prehypertension and hypertension consumed cigarettes on a daily basis and were physically active for less than 150 min.

Our results showed that the proportion of people with prehypertension was 1.5 times higher than the data obtained from the national survey conducted in 2013 [2].

For this reason, studies of this type are very important in order to identify vulnerable population groups and predictors of prehypertension and hypertension, with the aim of raising awareness for health and social policy makers so that they can formulate strategies that will intensify preventive activities related to this significant problem.

### Strengths and Limitations

Our study provided important data on risk factors associated with the high rates of prehypertension and hypertension. On the one hand, it would be beneficial to increase the public awareness of CVD risk factors for the purpose of its early detection and effective treatment of people with prehypertension so that hypertension could ultimately be prevented.

Regarding public health researchers, conducting such research is of great importance, given that it provides all the valuable data necessary for the creation and implementation of health education intervention strategies. With effective health and social policies and preventive interventions, it is possible not only to prevent or modify risk factors, but also to prevent complications, physical disabilities, and a lower quality of life, as well as premature death. Attention should be paid to the detection of unrecognized cases and people at risk by intensifying screening methods, which are otherwise carried out in Serbia within the primary health care framework, but to an insufficient extent.

The limitations of this study include the following: a cross-sectional research study design implies that no causal inference can be made as regards to the association between health behaviors and blood pressure values; the data based on the self-reporting of physical activity, cigarette and alcohol use could have introduced recall bias, which may have prevented us from accurately assessing the relationship between the variables and blood pressure categories.

## 5. Conclusions

Serbia belongs to the group of countries with a high prevalence of prehypertension and hypertension. Being of male sex, marriage, lower education, poverty, age and the presence of multimorbidity are significant predictors of prehypertension and hypertension in the Serbian adult population, so preventive strategies should be aimed at these sensitive population groups. In addition, it is necessary to prevent risk factors related to unhealthy lifestyle habits such as cigarette consumption and physical inactivity. Our results emphasize the need for a public health strategy for the prevention, detection and treatment of prehypertension and hypertension.

## Figures and Tables

**Table 1 medicina-60-00824-t001:** Demographic and sociomedical characteristics of the studied population in relation to blood pressure.

Variables		Study Population			*p*
		Normotension *n* = 1539	Prehypertension *n* = 5165	Hypertension *n* = 3811	
Gender	female	1141 (73.66%)	2550 (49.37%)	1824 (47.86%)	<0.001
	male	408 (26.34%)	2615 (50.63%)	1987 (52.14%)	
Age	20–29	335 (21.77%)	751 (14.54%)	95 (2.49%)	<0.001
	30–39	335 (21.77%)	878 (16.99%)	205 (5.37%)	
	40–49	296 (19.23%)	966 (18.70%)	414 (10.86%)	
	50–59	219 (14.23%)	946 (18.31%)	739 (19.39%)	
	60–69	189 (12.28%)	917 (17.75%)	1135 (29.78%)	
	70–79	122 (7.93%)	477 (9.23%)	823 (21.59%)	
	80+	53 (2.79%)	230 (4.48%)	400 (10.52%)	
Region	Belgrade	467 (30.34%)	922 (17.85%)	556 (14.58%)	<0.001
	Vojvodina	281 (18.26%)	1010 (19.55%)	995 (26.10%)	
	Šumadija and Western Serbia	532 (34.56%)	1985 (38.43%)	1236 (32.43%)	
	Southern and Eastern Serbia	269 (16.84%)	1248 (24.17%)	1024 (26.89%)	
Marital status	never married/unmarried community	386 (25.08%)	1043 (20.19%)	296 (7.76%)	<0.001
	divorce, separation, death of a partner	225 (14.62%)	795 (15.39%)	1005 (26.37%)	
	marriage/common-law union	935 (60.3%)	3320 (64.42%)	2506 (65.87%)	
Education	primary and lower school	274 (17.80%)	1089 (21.08%)	1390 (36.47%)	<0.001
	secondary school	860 (55.88%)	3054 (59.23%)	1933 (50.72%)	
	college and university	415 (26.32%)	1022 (19.69%)	487 (12.81%)	
Employment status	unemployed	312 (20.27%)	1047 (20.27%)	582 (15.27%)	<0.001
	inactive	542 (35.22%)	1854 (35.89%)	2279 (59.80%)	
	employed	694 (44.51%)	2264 (43.84%)	948 (24.93%)	
Well-being index	the poorest	610 (39.63%)	2075 (40.17%)	1735 (45.52%)	<0.001
	middle layer	312 (20.27%)	1030 (19.94%)	814 (21.35%)	
	the richest layer	627 (40,1%)	2060 (39,89%)	1262 (33.13%)	
Self-assessment of general health	bad and very bad	1101 (71.53%)	3564 (69%)	1677 (44%)	<0.001
	average	295 (19.16%)	1136 (21.99%)	1367 (35.86%)	
	good and very good	143 (9.31%)	465 (9.01%)	767 (20,14%)	
Multimorbidity	multimorbidity	401 (26.05%)	1479 (28.63%)	1990 (52.21%)	<0.001
	one disease	234 (15.20%)	919 (17.79%)	889 (23.32%)	
	no illness	904 (58.73%)	2767 (53.57%)	932 (24.45%)	
Depressiveness	no symptoms	24 (1.56%)	43 (0.83%)	53 (1.39%)	<0.001
	mild symptoms	1204 (78.23%)	4122 (79.80%)	2567 (67.35%)	
	a depressive episode	311 (20.21%)	1000 (19.37%)	1191 (31.25%)	
BMI	malnutrition	99 (6.43%)	140 (2.71%)	84 (2.20%)	<0.001
	normal nutrition	821 (53.34%)	2035 (39.39%)	1007 (26.42%)	
	preobesity	415 (26.96%)	2043 (39.55%)	1464 (38.41%)	
	obesity	204 (13.25%)	947 (18.33%)	1256 (32.95%)	

**Table 2 medicina-60-00824-t002:** Characteristics of the study population in relation to health behavior.

Vriables		Study Population			*p*
		Normotension *n* = 1539	Prehypertension *n* = 5165	Hypertension *n* = 3811	
Cigarette consumption	no answer	200 (12.99%)	899 (17.40%)	688 (18.05%)	<0.001
	yes, everyday	310 (20.14%)	1095 (21.20%)	792 (20.78%)	
	yes, occasionally	78 (5.07%)	198 (3.83%)	116 (3.04%)	
	not	824 (61.8%)	2696 (57.57%)	2013 (58.13%)	
Alcohol consumption	no answer	275 (17.86%)	1165 (22.12%)	989 (25.94%)	<0.001
	once a week or more often	189 (12.27%)	703 (13.59%)	579 (3.91%)	
	2–3 days a month and less often	578 (37.54%)	1805 (34.93%)	1141 (29.93%)	
	not, never	497 (32.29%)	1492 (28.88%)	1102 (28.91%)	
Fruit and vegetable consumption	no answer	23 (1.49%)	42 (0.81%)	52 (1.35%)	<0.001
	once or more times a day	622 (40.41%)	2029 (39.28%)	1430 (37.52%)	
	4–6 times a week	441 (28.65%)	1417 (27.43%)	954 (25.03%)	
	1–3 times a week	370 (24.04%)	1278 (24.74%)	1044 (27.39%)	
	less than once a week	67 (4.35%)	352 (6.81%)	277 (7.26%)	
	never	16 (1.04%)	47 (0.91%)	54 (1.41%)	
Physical activity	less than 150 min	1187 (77.12%)	4107 (79.51%)	3045 (79.90%)	<0.001
	more than 150 min	352 (22.88%)	1058 (20.49%)	766 (20.1%)	

**Table 3 medicina-60-00824-t003:** Regression analysis of the influence of demographic and sociomedical characteristics on the value of blood pressure in persons with prehypertension and hypertension.

Variables	Prehypertension				Hypertension			
	Univariate Model		Multivariate Model		Univariate Model		Multivariate Model	
	OR (95%)	*p*	OR (95%)	*p*	OR (95%)	*p*	OR (95%)	*p*
Gender								
female	0.328 (0.288–0.374)	<0.001	0.244 (0.211–0.283)	<0.001	0.349 (0.308–0.395)	<0.001	0.324 (0.284–0.370)	<0.001
male	1				1		1	
Age								
20–29	0.038 (0.026–0.054)		0.043 (0.027–0.067)	<0.001	0.517 (0.373–0.715)	<0.001	0.454 (0.305–0.674)	<0.001
30–39	0.081 (0.058–0.113)	<0.001	0.094 (0.062–0.142)	<0.001	0.604 (0.437–0.835)	0.002	0.527 (0.357–0.778)	<0.001
40–49	0.185 (0.134–0.256)	<0.001	0.202 (0.137–0.299)	<0.001	0.752 (0.543–1.042)	0.086	0.638 (0.434–0.937)	0.022
50–59	0.447 (0.323–0.618)	<0.001	0.507 (0.347–0.739)	<0.001	0.995 (0.714–1.389)	0.978	0.890 (0.610–1.299)	0.547
60–69	0.796 (0.575–1.102)	0.168	0.869 (0.614–1.230)	0.428	1.118 (0.798–1.566)	0.517	1.075 (0.754–1.534)	0.689
70–79	0.894 (0.634–1.261)	0.522	0.940 (0.659–1.341)	0.732	0.901 (0.629–1.290)	0.569	0.893 (0.618–1.289)	0.545
80+	1		1		1		1	
Region								
Belgrade	0.313 (0.261–0.375)	<0.001	0.338 (0.274–0.417)	<0.001	0.426 (0.358–0.506)	<0.001	0.399 (0.330–0.482)	<0.001
Vojvodina	0.930 (0.770–1.123)	0.452	0.976 (0.795–1.199)	0.816	0.775 (0.643–0.934)	0.007	0.755 (0.622–0.917)	0.005
Šumadija and Western Serbia	0.610 (0.516–0.722)	<0.001	0.630 (0.525–0.756)	<0.001	0.804 (0.683–0.946)	0.009	0.773 (0.653–0.915)	0.003
Southern and Eastern Serbia	1		1		1		1	
Marital status								
never married/unmarried community	0.286 (0.242–0.339)	<0.001	0.870 (0.697–1.085)	0.216	0.761 (0.663–0.873)	<0.001	0.904 (0.756–1.082)	0.272
divorce, separation, death of a partner	1.667 (1.416–1.961)	<0.001	1.136 (0.940–1.373)	0.188	0.995 (0.844–1.173)	0.953	1.050 (0.872–1.264)	0.604
marriage/common-law union	1		1		1		1	
Education								
primary and lower school	4.323 (3.596–5.197)	<0.001	2.103 (1.679–2.635)	<0.001	1.614 (1.355–1.922)	<0.001	1.348 (1.096–1.658)	<0.001
secondary school	1.915 (1.643–2.233)	<0.001	1.628 (1.364–1.943)	<0.001	1.442 (1.258–1.654)	<0.001	1.282 (1.103–1.489)	<0.001
college and university	1		1		1		1	
Employment status								
unemployed	0.687 (0.367–1.288)	0.242	1.167 (0.589–2.313)	0.657	0.824 (0.453–1.499)	0.526	1.208 (0.650–2.244)	0.550
inactive	1.549 (0.833–2.879)	0.166	1.071 (0.540–2.124)	0.845	0.840 (0.465–1.519)	0.564	1.038 (0.557–1.935)	0.907
employed	1		1		1		1	
Well-being index								
the poorest	1.413 (1.237–1.614)	<0.001	0.864 (0.731–1.021)	0.087	1.035 (0.912–1.176)	<0.001	0.762 (0.654–0.887)	0.592
middle layer	1.296 (1.102–1.524)	<0.001	0.968 (0.807–1.161)	0.726	1.005 (0.861–1.173)	0.066	0.856 (0.726–1.010)	0.952
the richest layer	1		1		1		1	
Self-assessment of general health								
bad and very bad	0.997 (0.815–1.220)	0.980	1.187 (0.937–1.504)	0.956	0.280 (0.230–0.341)	<0.001	0.622 (0.491–0.787)	<0.001
average	1.188 (0.944–1.495)	0.141	1.192 (0.932–1.524)	0.162	0.860 (0.690–1.073)	0.182	1.027 (0.807–1.307)	0.829
good and very good	1		1		1		1	
Multimorbidity								
multimorbidity	1.218 (1.066–1.392)	<0.001	1.505 (1.187–1.909)	0.001	4.867 (4.227–5.603)	<0.001	3.555 (2.758–4.582)	<0.001
one disease	1.297 (1.103–1.525)	<0.001	1.199 (0.901–1.595)	0.113	3.726 (3.142–4.418)	<0.001	2.401 (1.778–3.242)	<0.001
no illness	1		1		1		1	
Depressiveness								
no symptoms	0.557 (0.333–0.933)	0.026	0.563 (0.336–0.943)	0.029	0.577 (0.350–0.949)	0.030	0.611 (0.368–1.015)	0.057
mild symptoms	1.056 (0.916–1.218)	0.108	1.132 (0.973–1.317)	0.454	0.552 (0.479–0.637)	<0.001	0.849 (0.728–0.990)	0.037
a depressive episode	1		1		1		1	
BMI								
malnutrition	0.226 (0.161–0.319)	<0.001	0.334 (0.184–0.608)	<0.001	0.052 (0.033–0.083)	<0.001	0.100 (0.050–0.201)	<0.001
normal nutrition	0.509 (0.426–0.608)	<0.001	0.548 (0.416–0.722)	<0.001	0.182 (0.152–0.218)	<0.001	0.268 (0.204–0.352)	<0.001
preobesity	1.036 (0.855–1.255)	0.716	1.003 (0.750–1.342)	0.983	0.547 (0.452–0.662)	<0.001	0.625 (0.470–0.830)	<0.001
obesity	1		1		1		1	

**Table 4 medicina-60-00824-t004:** Regression analysis of the influence of health behaviors on the value of blood pressure in persons with prehypertension and hypertension.

Variables	Prehypertension				Hypertension			
	Univariate Model		Multivariate Model		Univariate Model		Multivariate Model	
	OR (95%)	*p*	OR (95%)	*p*	OR (95%)	*p*	OR (95%)	*p*
Cigarette consumption								
yes, everyday	1.080 (0.931–1.252)	0.311	1.062 (0.913–1.235)	0.435	1.046 (0.896–1.220)	0.570	1.113 (0.947–1.309)	0.193
yes, occasionally	0.776 (0.590–1.020)	0.069	0.762 (0.578–1.005)	0.162	0.609 (0.452–0.820)	0.001	0.670 (0.490–0.914)	0.012
not	1		1		1		1	
Alcohol consumption								
once a week or more often	1.604 (1.001–2.569)	0.049	1.304 (0.981–1.734)	0.067	2.740 (1.723–4.358)	0.001	2.815 (1.747–4.537)	0.001
2–3 days a month and less often	1.473 (1.006–2.157)	0.046	1.516 (1.033–2.266)	0.034	1.880 (1.280–2.762)	0.001	2.315 (1.558–3.440)	0.005
not	1		1		1		1	
Fruit and vegetable consumption								
Once or more times a day	1.093 (0.615–1.941)	0.762	1.138 (0.634–2.044)	0.664	0.670 (0.381–1.180)	0.166	0.747 (0.419–1.334)	0.325
4–6 times a week	1.094 (0.614–1.948)	0.761	1.114 (0.618–2.007)	0.720	0.641 (0.363–1.132)	0.126	0.701 (0.392–1.256)	0.223
1–3 times a week	1.176 (0.659–2.098)	0.583	1.187 (0.658–2.140)	0.570	0.836 (0.473–1.479)	0.538	0.849 (0.474–1.522)	0.583
less than once a week	1.789 (0.958–3.340)	0.068	1.785 (0.949–3.357)	0.072	1.225 (0.660–2.274)	0.520	1.245 (0.665–2.331)	0.494
not, never	1		1		1		1	
Physical activity								
less than 150 min	1.213 (1.031–1.427)	0.020	1.168 (0.857–1.591)	0.325	3.437 (2.802–4.215)	0.001	3.166 (2.176–4.605)	0.001
more than 150 min	1		1		1		1	

## Data Availability

Data are unavailable due to privacy or ethical restrictions because the current owner of the rights, the Institute of Public Health of Serbia, “Milan Jovanović Batut” and the database was handed over to the University of Kragujevac with an official letter for the purpose of further research.

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
