# Peer review of "Demographic and Socioeconomic Predictors of Prehypertension and Hypertension in the Adult Population: Serbian National Health Survey"

_medicina, 2024, doi:10.3390/medicina60050824_

Round 1

Reviewer 1 Report

Comments and Suggestions for Authors

The objective of this study were to determine the frequency and demographic and socioeconomic predictors of prehypertension and hypertension in the adults in Serbia, and to examine the relationship between prehypertension and hypertension and determinants health behavior (smoking, alcohol use, physical activity) and individual aspects of health (self-assessment of health, multimorbidity, BMI, depressive symptoms). The manuscript includes interesting results but it needs major modifications. Further, the similarity report showed high percentage of similarity mainly self-plagiarism in methodology section which needs to be re-phrased.

Abstract need to be re-written for clarity mainly sub-sections of Methodology and Background and objective

L27-29: the objective is not mentioned in the abstract section background and objective sub-section

L31-32 delete “"Dr. Milan Jovanović Batut" no need to add this information

L34: “were” instead of “will be”

After L60: add some information about the factors affecting the pre-hypertension and hypertension

L70: “Serbia” instead of “our country”

L70-72: re-write for clarity

L127-131: a copy of questionnaire could be provided as supplementary materials

It seems that the X2 values in Tables 1 and 2 are for the differences in the answers of each question among each group. Please make sure that the statistical analysis between the tested group.

It is better to combine the results and discussion in one section to avoid repeating the results in the section of discussion
the discussion should be elaborated and more references should be used to include more recent studies mainly in EU and the factors associated with prehypertension and hypertension should be well discussed.  

A section of strengths and limitation should be added before the conclusions

The conclusion section contains only general information therefore, it should be deleted and re-written in good way to show the most important results and their significance and some future studies and recommendation.

Comments on the Quality of English Language

Moderate editing of English language required

Author Response

First of all, we would like to thank you for taking the time to review our work. We tried to respect your comments and to correct the observed irregularities that you pointed out. We have marked corrections in manuscript.

The objective of this study were to determine the frequency and demographic and socioeconomic predictors of prehypertension and hypertension in the adults in Serbia, and to examine the relationship between prehypertension and hypertension and determinants health behavior (smoking, alcohol use, physical activity) and individual aspects of health (self-assessment of health, multimorbidity, BMI, depressive symptoms). The manuscript includes interesting results but it needs major modifications. Further, the similarity report showed high percentage of similarity mainly self-plagiarism in methodology section which needs to be re-phrased.

Abstract need to be re-written for clarity mainly sub-sections of Methodology and Background and objective

L27-29: the objective is not mentioned in the abstract section background and objective sub-section

We corrected.

L31-32 delete “"Dr. Milan Jovanović Batut" no need to add this information

We corrected.

L34: “were” instead of “will be”

We corrected.

After L60: add some information about the factors affecting the pre-hypertension and hypertension

We corrected.

L70: “Serbia” instead of “our country”

We corrected.

L70-72: re-write for clarity

We corrected.

L127-131: a copy of questionnaire could be provided as supplementary materials.

We corrected.

It seems that the X2 values in Tables 1 and 2 are for the differences in the answers of each question among each group. Please make sure that the statistical analysis between the tested group.

We checked the statistical analysis.

It is better to combine the results and discussion in one section to avoid repeating the results in the section of discussion the discussion should be elaborated and more references should be used to include more recent studies mainly in EU and the factors associated with prehypertension and hypertension should be well discussed.  

We combined results and discussion.

A section of strengths and limitation should be added before the conclusions.

We've added strengths and limitations.

Our study provided important data on risk factors associated with the high rates of pre-hypertension and hypertension. On the one hand, it would be beneficial to increase public awareness of CVD risk factors for the purpose of its early detection and effective treatment of people with prehypertension so that hypertension could ultimately be prevented.

As  public health researchers, conducting such research is of great importance, given that it provides all the valuable data necessary for the creation and implementation of health education intervention strategies. With effective health and social policies and preventive interventions, it is possible not only to prevent or modify risk factors, but also to prevent complications, physical disabilities, a lower quality of life, as well as premature death. Attention should be paid to the detection of unrecognized cases and people at risk by intensifying screening methods, which are otherwise carried out in Serbia within the primary health care framework, but to an insufficient extent.

Limitations of this study include the following: a cross-sectional research study design implies that no causal inference can be made as regards the association between health behaviors and blood pressure values; the data based on self-report of physical ac-tivity, cigarette and alcohol use could have introduced recall bias, which may have pre-vented us from accurately assessing the relationship between the variables and blood pressure categories.

The conclusion section contains only general information therefore, it should be deleted and re-written in good way to show the most important results and their significance and some future studies and recommendation.

We corrected.

Serbia belongs to countries with a high prevalence of prehypertension and hypertension. Male sex, marriage, lower education, poverty, age and the presence of multimorbidity are significant predictors of prehypertension and hypertension in the Serbian adult population, so preventive strategies should be aimed at these sensitive population groups. In addition, it is necessary to prevent risk factors related to unhealthy lifestyle habits such as cigarette consumption and physical inactivity. Our results emphasize the need for a public health strategy for the prevention, detection and treatment of prehypertension and hypertension.

Moderate editing of English language required.

We corrected.

Reviewer 2 Report

Comments and Suggestions for Authors

 Demographic And Socioeconomic Predictors of Prehypertension and Hypertension in The Adult Population: Serbian National Health Survey

Dimitrijev etal

I would like to thank the authors and the Editors for allowing me the opportunity to evaluate their work. The authors utilized the fourth National Population Health Survey conducted in 2019 to assess the frequency and demographic and socioeconomic predictors of prehypertension and hypertension in the Republic of Serbia. The topic is important. Although the authors exerted much effort on this work, the manuscript in it is current situation lacks important information regarding the methodology of the statistical analysis, which affects the overall study outcome. My concerns are the following

The abstract lacks essential information like the objectives and the main research outcome. The results writing also need improvement: “For example, lines 36&37 age 50-59 years old, have a 0.4 times higher risk for prehypertension (O = 0.447), actually based on the OR, they are at reduced risk by 60%. The writing of the introduction needs improvement; some of its selections are difficult to follow, for example, lines 62 and 63. The survey sampling method was a stratified two-stage cluster sample, while the statistical analysis is based on simple random sampling. What is the implication of this complex sampling design on the statistical analysis, this is unclear? The effects of complex sampling designs on statistical analysis necessitate careful consideration and appropriate adjustment to ensure accurate and reliable estimation of population parameters. Failure to account for the complexities introduced by the sampling design can lead to biased results and erroneous conclusions. Moreover authors should calculate and report effect size measures. P-values are highly correlated with the size of the sample. The authors present results on a fairly large sample size; therefore, most of the tested differences would be significant by default. The presentation of tables in the Results section needs improvement, and the interpretation of the OR needs to be revised with more accuracy  

Author Response

I would like to thank the authors and the Editors for allowing me the opportunity to evaluate their work. The authors utilized the fourth National Population Health Survey conducted in 2019 to assess the frequency and demographic and socioeconomic predictors of prehypertension and hypertension in the Republic of Serbia. The topic is important. Although the authors exerted much effort on this work, the manuscript in it is current situation lacks important information regarding the methodology of the statistical analysis, which affects the overall study outcome. My concerns are the following

The abstract lacks essential information like the objectives and the main research outcome. The results writing also need improvement: “For example, lines 36&37 age 50-59 years old, have a 0.4 times higher risk for prehypertension (O = 0.447), actually based on the OR, they are at reduced risk by 60%. The writing of the introduction needs improvement; some of its selections are difficult to follow, for example, lines 62 and 63. The survey sampling method was a stratified two-stage cluster sample, while the statistical analysis is based on simple random sampling. What is the implication of this complex sampling design on the statistical analysis, this is unclear? The effects of complex sampling designs on statistical analysis necessitate careful consideration and appropriate adjustment to ensure accurate and reliable estimation of population parameters. Failure to account for the complexities introduced by the sampling design can lead to biased results and erroneous conclusions. Moreover authors should calculate and report effect size measures. P-values are highly correlated with the size of the sample. The authors present results on a fairly large sample size; therefore, most of the tested differences would be significant by default. The presentation of tables in the Results section needs improvement, and the interpretation of the OR needs to be revised with more accuracy  

Answer

First of all, we would like to thank you for taking the time to review our work. We tried to respect your comments and to correct the observed irregularities that you pointed out. We have marked corrections in manuscript.

The authors thank the reviewer for pointing out an error in the interpretation of the results.

We have now listed the corrected data in the summary and results section related to the regression analysis so that it reads:

Abstract: Background and Objectives: Prehypertension and hypertension are the most common cardiovascular disorders worldwide and are increasingly considered one of the most serious public health problems, especially in developing countries. Materials and Methods: The research is part of the fourth National Population Health Survey conducted in 2019, which was conducted by the Republic Institute of Statistics, in cooperation with the Institute for Public Health of Serbia "Dr. Milan Jovanović Batut" and the Ministry of Health of the Republic of Serbia. As a research instrument, questionnaires were used in accordance with the methodology of the European Health Survey. For the purposes of this research, data on the adult population aged 20 and over will be used. Results: Women are at reduced risk for both prehypertension (OR = 0.328) and hypertension (OR=0.349) by nearly 70%. Similar, age below 60 years have lower risk for prehypertension and younger than 40 years have loer risk for hypertension (OR=0.995), while people with lower education have 4.3 times higher risk of prehypertension (OR= 4.323) and 1.6 times of hypertension (OR= 1.614). The poor have a 1.4 times higher risk of prehypertension (OR=1.413) and 1 times of hypertension (OR=1.035). People with multimorbidity have a 1.2 times higher risk of both prehypertension (OR=1.218) and 4.8 times of hypertension (OR=4.867). Conclusions: With effective preventive interventions, it is possible to prevent or modify risk factors, prevent complications, disability, reduced quality of life, and premature death in adults with prehypertension and a high risk of developing hypertension.

Results section:

…. Tables 3 and 4 show the predictive significance of the respective socio-demographic characteristics, health status and health behavior of the respondents for the occurrence of prehypertension and hypertension. Women are at reduced risk to have prehypertension (OR = 0.328) i.e. hypertension (OR=0.349) by nearly 70%. When it comes to marital status, it can conclude that divorced people are 1.6 times more likely to have prehypertension (OR=1.667), while never married subjects have reduced risk for both prehypertension (OR= 0.286) and hypertension (OR=0.761) by 70% i.e. 20%. All age groups under 60 years have a significantly lower risk of prehypertension and those younger than 40 years of hypertension compared to the oldest population, residents of Šumadija and western Serbia reduced risk of prehypertension (OR=0.610) by 40% and reduced risk for hypertension (OR=0.804) by 20%, while people with lower education have 4.3 times higher risk of prehypertension (OR= 4.323) and 1.6 times of hypertension (OR= 1.614)….

Reviewer 3 Report

Comments and Suggestions for Authors

The results section requires extensive revision  as it has serious flaws. Basic statistical interpretation errors.

OR is not properly interpreted.

Authors are not taking into account p values and multivariate model for the claims on their results 

Comments on the Quality of English Language

need to be revised

Author Response

The results section requires extensive revision  as it has serious flaws. Basic statistical interpretation errors.

OR is not properly interpreted.

Authors are not taking into account p values and multivariate model for the claims on their results 

Answer

First of all, we would like to thank you for taking the time to review our work. We tried to respect your comments and to correct the observed irregularities that you pointed out. We have marked corrections in manuscript.

The authors thank the reviewer for pointing out an error in the interpretation of the results.

We have now listed the corrected data in the summary and results section related to the regression analysis so that it reads:

common cardiovascular disorders worldwide and are increasingly considered one of the most serious public health problems, especially in developing countries. Materials and Methods: The research is part of the fourth National Population Health Survey conducted in 2019, which was conducted by the Republic Institute of Statistics, in cooperation with the Institute for Public Health of Serbia "Dr. Milan Jovanović Batut" and the Ministry of Health of the Republic of Serbia. As a research instrument, questionnaires were used in accordance with the methodology of the European Health Survey. For the purposes of this research, data on the adult population aged 20 and over will be used. Results: Women are at reduced risk for both prehypertension (OR = 0.328) and hypertension (OR=0.349) by nearly 70%. Similar, age below 60 years have lower risk for prehypertension and younger than 40 years have loer risk for hypertension (OR=0.995), while people with lower education have 4.3 times higher risk of prehypertension (OR= 4.323) and 1.6 times of hypertension (OR= 1.614). The poor have a 1.4 times higher risk of prehypertension (OR=1.413) and 1 times of hypertension (OR=1.035). People with multimorbidity have a 1.2 times higher risk of both prehypertension (OR=1.218) and 4.8 times of hypertension (OR=4.867). Conclusions: With effective preventive interventions, it is possible to prevent or modify risk factors, prevent complications, disability, reduced quality of life, and premature death in adults with prehypertension and a high risk of developing hypertension.

Results section:

…. Tables 3 and 4 show the predictive significance of the respective socio-demographic characteristics, health status and health behavior of the respondents for the occurrence of prehypertension and hypertension. Women are at reduced risk to have prehypertension (OR = 0.328) i.e. hypertension (OR=0.349) by nearly 70%. When it comes to marital status, it can conclude that divorced people are 1.6 times more likely to have prehypertension (OR=1.667), while never married subjects have reduced risk for both prehypertension (OR= 0.286) and hypertension (OR=0.761) by 70% i.e. 20%. All age groups under 60 years have a significantly lower risk of prehypertension and those younger than 40 years of hypertension compared to the oldest population, residents of Šumadija and western Serbia reduced risk of prehypertension (OR=0.610) by 40% and reduced risk for hypertension (OR=0.804) by 20%, while people with lower education have 4.3 times higher risk of prehypertension (OR= 4.323) and 1.6 times of hypertension (OR= 1.614)….

Round 2

Reviewer 1 Report

Comments and Suggestions for Authors

The manuscript is improved and can be accepted in the current form

Author Response

The authors thank the reviewer.

Reviewer 2 Report

Comments and Suggestions for Authors

Thank you for revising the manuscript and responding to our inquiries; however, we still require clarification regarding the following.

-        The survey sampling method was a stratified two-stage cluster sample [ lines 107-110], while the statistical analysis is based on simple random sampling. What is the implication of this complex sampling design on the statistical analysis? This is unclear. Please clarify

-        Failure to consider the intricacies of sample design might result in biased results and incorrect conclusions.

-        Authors should calculate and report effect size measures. P-values are highly correlated with the size of the sample. The authors present results on a fairly large sample size; therefore, most of the tested differences would be significant by default.

-        The authors have combined the results and discussion sections in their manuscript [ lines 194-286] , which has made it quite challenging to follow. Typically, the results section presents the study's findings in a clear and concise manner without interpreting or discussing their implications. The separation of these two sections enables the reader to more easily comprehend the primary research findings and their interpretations and implications.

-        Still, some errors are found, for example, in line 41 “have loer risk”. Moreover, there was no consistency in writing the p-values ( 0,001 and 0.001), and the same applies to the OR and percentages. Also, it is written in the column χ2, while it reported the p-value, not the χ2 statistic value (Table 1).

Author Response

Thank you for your efforts to improve our manuscript. We've done our best to clarify any ambiguities and appreciate your comments. We marked the corrections in blue in manuscript.

-        The survey sampling method was a stratified two-stage cluster sample [ lines 107-110], while the statistical analysis is based on simple random sampling. What is the implication of this complex sampling design on the statistical analysis? This is unclear. Please clarify

-        Failure to consider the intricacies of sample design might result in biased results and incorrect conclusions.

The sampling methodology was done according to the European Health Interview Survey (EHIS wave 3) - Methodological manual, Eurostat, 2018.

European Health Interview Survey European Health Interview Survey (EHIS) represents a periodic survey on the health of the population, with the use of widely accepted standardized instruments (interview survey), to gather reliable data about the health condition, health protection and determinants of health of the population of the European Union (EU). It also represents a representative survey for calculating a large number of European Core Health Indicators, 1 known as ECHI indicators, which enable the comparison of population health among European countries, both in the year of the research and over time. The first wave of EHIS was conducted in 17 EU member states in the period from 2006 to 2009. The second wave of EHIS was conducted in all 28 EU member states, as well as in Iceland, Norway and Turkey between 2013 and 2015. Work on the third wave of EHIS was launched at the European Union Statistical Office (EUROSTAT) in 2016. During the period from 2016 to 2018, detailed discussions and consultations were organized by various bodies of the European Statistical System (ESS) Task Force, the Technical Group for Health Research, the Working Group on Public Health Statistics and the Group of Directors of Social Statistics - National statistical institutes. Regulations on the implementation of the third wave of EHIS were made by the European Commission in 2018 as the Commission Regulation for Implementation (EU) No. 255/20184.2 Serbian National Health Survey The 2019 Serbian National Health Survey is the fourth national health survey of the Serbian population using the European Health Interview Survey instruments, conducted by the Statistical Office of the Republic of Serbia in cooperation with the Institute of Public Health of Serbia and the Ministry of Health of the Republic of Serbia in 2019. In the Republic of Serbia, the National Health Survey was conducted for the first time in 2000 with the financial and technical support of the World Health Organization and UNICEF. Research under the title "Health status, health needs and utilization of health services in Serbia", was conducted by the Institute of Public Health of Serbia in cooperation with a network of public health institutes and primary health care facilities. The results of the research represented one of the baselines in creation of the health policies in Serbia in 2002. Further research was conducted in 2006, funded by the loan of the World Bank, which the Republic of Serbia received for the implementation of the project "Development of Health of Serbia". In this research the protocol and the survey from 2000 represented the starting material on which improvements and additions were made. It was made sure not to jeopardize the comparability with the findings of previous research, but was made possible to obtain answers to standardized questions, which would be used in research in the EU, concluding with the assessment of health indicators recommended by the World Health Organization3 and the European Commission. 4 The third Serbian National Health Survey was conducted in 2013 from the funds of the project "Decentralization of services at the local level" (DILS), financed from a loan from the World Bank. The research included additional harmonization of research instruments (methodology, questionnaires, and instructions) with the European Health Interview Survey instruments - the second wave (EHIS - wave 2). A step forward was made when the Statistical Office of the Republic of Serbia, in accordance with the process of accession of the Republic of Serbia to the EU and the National Program for the Adoption of the EU Acquis, included statistical data obtained by conducting the European Health Interview Survey in the Official Statistics Program for 2016-2020. In research conducted in 2019, a new harmonization of research instruments was performed with the instruments of the European Health Interview Survey - the third wave (EHIS - wave 3).5 Financial support was provided by the Government of the Republic of Serbia and the Instrument for Pre-Accession Assistance (IPA) 2018.

Sample plan

In accordance with the recommendations for conducting the European health Interview Survey,a sample plan was defined. The sample plan includes the planned sample size, sample selection framework, domain selection, sample allocation, sample selection stages, stratification, and sample weight calculation.

Target population

The primary target population consisted of all persons aged  and over living in private (non-institutional) households in the Republic of Serbia, who represent the usual population. Excluded are persons in collective households (studentdormitories, dormitories for children and youth with disabilities, homes for socially endangered children, homes for pensioners, the elderly and infirm, homes for adults with disabilities, monasteries, convents, etc.).The special target population consisted of children aged 5 to 14 years.

Domains of research and stratification

The main goal of the research was to obtain statistically reliable estimates for most indicators, both at the national level and for urban and other settlements, as well as for four regions: Belgrade region, Vojvodina region, Sumadija and Western Serbia region, Southern and Eastern Serbia region. In the research, a stratified two-stage sample was applied, which is stratified according to settlements and regions (Region of Šumadija and West Serbia, Region of South and East Serbia, Region of Vojvodina and Belgrade Region). The data of the 2011 population census,  were used as a framework for selecting the sample.

Sample size and sample allocation

The sample size was calculated on the basis of the requirements related to the precision of estimates, in order to assess the standard error of the indicator “proportion of persons prevented from engaging in daily activities” in accordance with EUROSTAT recommendations for conducting the European Health Interview Survey [13]. In EU countries, based on the SILK survey, the proportion of people who are prevented from performing daily activities in the population varies between 4% and 11%. Similar results were obtained in the SILK survey in the Republic of Serbia (5–8%). If 8% is taken as the basis for further calculations, in order to estimate the parameter with an error of less than 1%, it is necessary for a simple random sample to have about 6,000 subjects aged 15 and over. In that case, if the statistic has a value of 8%, the parameter estimate is in the interval between 7.3% and 8.7% (95% confidence interval). It is planned to obtain statistically reliable estimates at the level of Serbia as a whole, then at the level of four regions: Belgrade region, Vojvodina region, Sumadija and Western Serbia region, Southern and Eastern Serbia region, as well as for the population of urban and other settlements. As a compromise between the required precision of estimates and the cost of conducting the research, a sample size of 6,000 households was determined, in which approximately 15,000 members were aged 15 and over whereas about 1,500 children were aged from 5 to 14. When calculating the sample size, children aged from 5 to 14 years were not taken into account. It was determined that 10 households should be selected in each census round, taking into account the costs of conducting the survey, as well as the time required to complete the survey in the census round. Reserve households were also provided for each census round in the case that a large number of households in the census round refused to cooperate. Dividing the total number of households by the number of households in the sample per census district, it was calculated that 600 census districts should be selected. Sample allocation by region and type of settlement, was proportional to the number of persons aged 15 and over in these contingents, based on the current 2018 demographic estimates. A sample of 5,114 households counted 15,621 persons, out of which there were 13,589 persons aged 15 and over and 1,493 children aged 5 to 14. For the purpose of this study, data on the adult population aged 20 years and older were used.

Sample selection frame and sample selection

The 2011 census of the Republic of Serbia was used as a framework for sampling. Census circles formed for the purposes of conducting the census, where they were defined as primary sample units and were selected from each stratum systematically with a probability of choice proportional to size, and the size measure was the number of households in each census circle based on the 2011 Census. Within each stratum, the census districts are sorted according to the municipality to which they belong and the ordinal number within the municipality. In this way, with systematic selection, a high level of implicit geographical stratification and effective sample distribution is ensured. Households within each census round were selected with equal probability (simple random) from the list of households recorded in the 2011 Census.

Calculation of weights

In order for the estimates from the sample to be representative of the observed general population, each household and person from the sample was assigned a weight. The main component of the weight is the reciprocal value of the product of the probability of selection in each stage for each stratum and represents the basic weight (weight of the sample plan). The second weight component takes into account the level of non-response for the household. Upon completion of fieldwork, response rates were calculated for each stratum. They were used to correct the sample plan weights calculated for each census round. The final weight for the person and the household was calculated on the basis of the adjusted weight for the household, using the calibration method. Adjusted household weights are multiplied by calibration factors so that population estimates are obtained in accordance with current demographic estimates for 2019. The conditions that were met when calculating the calibration factors are as follows: population distribution by sex (two groups) and by five years of age (16 groups), at the regional level (NSTJ level 2) and household distribution by number of household members (six groups), provided that the household and each person from the observed household have the same final weight, which provides consistent estimates on a household and person basis. Sample weights were associated with all databases and analyses were performed by weighting the data for each household and individual.

Respondents’ participation in the research and response rate

A total of 5,114 households and 15,621 respondents were included in the survey. Out of a total of 6,335 contacted households, 5,114 of them agreed to participate in the research. The response rate of the households was 80.7%. Out of a total of 13,589 registered members of households aged 15 and over, 13,178 of them agreed or was able to be surveyed, which gives a response rate of 97.0%. Out of the number of people who agreed to be interviewed, 11,790 agreed to fill in the self-complete questionnaire (response rate 89.5%), while 11,474 agreed to have their measurements taken (response rate 87.1%).

-        Authors should calculate and report effect size measures. P-values are highly correlated with the size of the sample. The authors present results on a fairly large sample size; therefore, most of the tested differences would be significant by default.

It is a national health survey conducted on a large sample of the population. The methodology is aligned with the European third wave health research.

-        The authors have combined the results and discussion sections in their manuscript [ lines 194-286] , which has made it quite challenging to follow. Typically, the results section presents the study's findings in a clear and concise manner without interpreting or discussing their implications. The separation of these two sections enables the reader to more easily comprehend the primary research findings and their interpretations and implications.

The original version of the manuscript we submitted had a separate results and discussion section. However, one of the reviewers suggested that we merge these two sections, and that is the reason for this type of structure. We can restore it to its original form if the editor or reviewers request it.

 -        Still, some errors are found, for example, in line 41 “have loer risk”. Moreover, there was no consistency in writing the p-values ( 0,001 and 0.001), and the same applies to the OR and percentages. Also, it is written in the column χ2, while it reported the p-value, not the χ2 statistic value (Table 1).

We corrected.

Reviewer 3 Report

Comments and Suggestions for Authors

I found odd the structure of the manuscript. Putting together the results and discussion sections. I am unaware of the journal's format but I haven't seen an structure like this anywhere before.

Normally the results will underline the numerical findings, highlighting the significative ones while the discussion is used to elaborate possible reasons for the results, and add supporting and / or contrasting data from the literature

The multivariable model and significance (A p-value < 0.05 is considered statistically significant ) are not taken into account when detailing the results, hence some of the asseverations need to be revist

Comments on the Quality of English Language

some typos detected

Author Response

Thank you for your efforts to improve our manuscript. We've done our best to clarify any ambiguities and appreciate your comments.

I found odd the structure of the manuscript. Putting together the results and discussion sections. I am unaware of the journal's format but I haven't seen an structure like this anywhere before.

Normally the results will underline the numerical findings, highlighting the significative ones while the discussion is used to elaborate possible reasons for the results, and add supporting and / or contrasting data from the literature

The original version of the manuscript we submitted had a separate results and discussion section. However, one of the reviewers suggested that we merge these two sections, and that is the reason for this type of structure. We can restore it to its original form if the editor or reviewers request it.

The multivariable model and significance (A p-value < 0.05 is considered statistically significant ) are not taken into account when detailing the results, hence some of the asseverations need to be revist

Limitations in the number of words limited us in the details of the results

Round 3

Reviewer 2 Report

Comments and Suggestions for Authors

Thank you, authors, for responding to the queries sent earlier. However, they did not answer the questions about the statistical analysis.   

The authors argued that the survey is a national health survey conducted on a large population sample, and the methodology is aligned with the European third-wave health research. I am not asking about the survey's methodology; it is perfect; my questions were straightforward about the statistical analysis conducted by the study team, where the statistical analysis conducted by the study team did not consider the sampling design's complexity and the use of weights.

Author Response

Dear Reviewer,

Thank you very much for taking the time to read and analyze our manuscript. We are very grateful for your detailed response. Your comments and expertise were very valuable to us and we honestly think you helped us make improvements to our manuscript.

Reviewer(s)' Comments to Author:

  • Thank you, authors, for responding to the queries sent earlier. However, they did not answer the questions about the statistical analysis.  The authors argued that the survey is a national health survey conducted on a large population sample, and the methodology is aligned with the European third-wave health research. I am not asking about the survey's methodology; it is perfect; my questions were straightforward about the statistical analysis conducted by the study team, where the statistical analysis conducted by the study team did not consider the sampling design's complexity and the use of weights.

Answer:

Calculation of weights

In order for the estimates from the sample to be representative of the observed general population, each household and person from the sample was assigned a weight. The main component of the weight is the reciprocal value of the product of the probability of selection in each stage for each stratum and represents the basic weight (weight of the sample plan). The second weight component takes into account the level of non-response for the household. Upon completion of fieldwork, response rates were calculated for each stratum. They were used to correct the sample plan weights calculated for each census round. The final weight for the person and the household was calculated on the basis of the adjusted weight for the household, using the calibration method. Adjusted household weights are multiplied by calibration factors so that population estimates are obtained in accordance with current demographic estimates for 2019. The conditions that were met when calculating the calibration factors are as follows: population distribution by sex (two groups) and by five years of age (16 groups), at the regional level (NSTJ level 2) and household distribution by number of household members (six groups), provided that the household and each person from the observed household have the same final weight, which provides consistent estimates on a household and person basis. Sample weights were associated with all databases and analyses were performed by weighting the data for each household and individual.

We can add this paragraph to the manuscript if necessary.

Reviewer 3 Report

Comments and Suggestions for Authors

My only comment is that the results and conclusions are merged together in the same section which is not the typical presentation for a paper. If it is the journal's format it is ok then.

Comments on the Quality of English Language

minor typos

Author Response

Dear Reviewer,

Thank you very much for taking the time to read and analyze our manuscript. We are very grateful for your detailed response. Your comments and expertise were very valuable to us and we honestly think you helped us make improvements to our manuscript.

Reviewer(s)' Comments to Author:

My only comment is that the results and conclusions are merged together in the same section which is not the typical presentation for a paper. If it is the journal's format it is ok then.

Answer: The original version of the manuscript we submitted had a separate Results and Discussion section. However, one of the reviewers suggested that we merge these two sections and that is the reason for this type of structure. We can return it to its original form if this structure is not in accordance with the journal editorial policy.
